# Bone-Targeting Radionuclides in the Treatment of Metastatic Castration-Resistant Prostate Cancer: A Review on Radium-223 Chloride (Alpharadin) in Combination with Other Therapies

**DOI:** 10.3390/diagnostics14212407

**Published:** 2024-10-29

**Authors:** Ali H. D. Alshehri

**Affiliations:** Department of Radiological Sciences, Faculty of Applied Medical Sciences, Najran University, Najran 61441, Saudi Arabia; ahzafer@nu.edu.sa

**Keywords:** prostate cancer, bone metastasis, radium-223 therapy

## Abstract

Recent advances have broadened the range of therapeutic options for mCRPC, with several new treatments, including novel hormonal therapies (enzalutamide, abiraterone), chemotherapeutic agents (docetaxel, cabazitaxel), immunotherapies (sipuleucel-T), and bone targeting radiopharmaceuticals (radium-223) showing improved clinical outcomes and receiving U.S. Food and Drug Administration approval. These new treatments provide new avenues for improving patient survival and quality of life. Radium-223, a targeted alpha-emitter, specifically targets bone metastases, offering palliative benefits and a potential increase in life expectancy. The integration of radium-223 with other treatments shows promise for managing mCRPC. However, the optimal sequencing and combination of radium-223 with other therapies are still being explored, with various clinical trials investigating new therapeutic approaches. The integration of these therapies, especially to provide more effective, personalized treatment strategies, requires further investigation. A thorough literature review was conducted on current treatments for mCRPC, including chemotherapeutic agents, oral hormonal therapies targeting the androgen receptor axis, immunotherapies, and radium-223. Ongoing clinical trials investigating radium-233 in the context of other therapies for the treatment of mCRPC patients were also reviewed. Further studies should focus on determining the optimal sequencing and dosing and identifying biomarkers that predict treatment response to enhance outcomes of mCRPC patients. This review underlines the rational strategies of combining radium-223 with other therapies, investigating their impact on bone in terms of delaying skeletal-related events, and managing bone disease progression in mCRPC patients.

## 1. Introduction

### 1.1. Prostate Cancer as a Global Health Challenge

Prostate cancer is one of the most significant health challenges worldwide, affecting millions of men annually. As the second most common cancer among men, it accounts for substantial morbidity and mortality. The burden of prostate cancer is expected to rise with the aging global population, necessitating an increased focus on its management and treatment. Notable geographical disparities in prostate cancer incidence and outcomes have been observed, with higher rates in developed regions such as North America and Europe compared with Asia and Africa. Such variations are due to differences in genetic predisposition, lifestyle factors, healthcare infrastructure, and screening practices [1,2].

Geographical disparities in incidence and mortality are significant, influenced by factors such as access to healthcare services, socioeconomic status, and public health policies. Developed countries with more advanced healthcare systems and better access to diagnostic tools, innovative treatments, and comprehensive cancer care programs report higher incidence rates but lower mortality rates. In contrast, people in developing regions face delayed diagnoses and poorer outcomes because of limited resources, lack of awareness, and insufficient healthcare infrastructure. In addition, cultural attitudes toward health and medical intervention, as well as the prevalence of risk factors, such as diet and lifestyle, contribute to these disparities. Understanding these geographical differences is crucial for tailoring public health strategies and improving global prostate cancer outcomes [1,2].

The progression of the disease from localized to advanced stages poses significant clinical challenges. Advanced prostate cancer, particularly metastatic castration-resistant prostate cancer (mCRPC), is difficult to treat and is associated with poor prognosis. mCRPC develops when prostate cancer continues to progress despite androgen deprivation therapy (ADT), highlighting the need for novel therapeutic approaches [3,4,5].

In recent years, new therapeutic agents, such as androgen receptor signaling inhibitors, chemotherapy, immunotherapy, and radiopharmaceuticals, have expanded the treatment landscape. Radium-223, a targeted alpha-emitter, has emerged as a promising option for mCRPC, offering survival benefits and improved quality of life. However, optimization of its use with other therapies remains a critical area of ongoing research [3,4,5].

### 1.2. Significance of mCRPC

mCRPC represents a critical and formidable phase in the continuum of prostate cancer progression. The clinical significance of mCRPC lies in its complex biology, aggressive nature, and limited treatment options, all of which contribute to a high mortality rate and substantial patient morbidity. It typically emerges after the failure of initial treatments and surgery, radiation, and ADT to contain it. The transition to castration resistance signifies that prostate cancer cells have adapted to low testosterone environments and continue to proliferate through alternative androgen receptor signaling pathways and other survival mechanisms. This adaptability makes mCRPC a particularly challenging condition to manage, as traditional hormone therapies lose their efficacy. The impact of mCRPC is profound, not only because of the aggressive disease course but also because of its associated symptoms, which significantly impair quality of life. Bone metastases are common in mCRPC, which can lead to severe bone pain, fractures, and other skeletal-related events. These complications necessitate a multidisciplinary approach to management, including pain control, bone health maintenance, and systemic cancer therapies [6,7,8].

The clinical challenges posed by advanced prostate cancer, particularly mCRPC, significantly impact patients’ quality of life and present a considerable economic burden, emphasizing the need for cost-effective management strategies and the development of novel therapies. Recent advances in mCRPC treatment, including novel androgen receptor inhibitors, chemotherapeutics, immunotherapies, and radiopharmaceuticals like radium-223, provide new avenues for improving patient survival and quality of life. Radium-223 specifically targets bone metastases, offering palliative benefits and a potential increase in life expectancy. However, the optimal integration of these therapies, especially to provide more effective, personalized treatment strategies, requires further investigation [9].

### 1.3. Overview of Current Treatments and Recent Advances

ADT remains the cornerstone of the management of advanced prostate cancer. It aims to reduce androgen levels that fuel cancer growth, typically through surgical castration or pharmaceutical agents. Although ADT initially results in tumor regression and symptomatic relief, its effectiveness is temporary, as nearly all patients eventually develop CRPC. The transition to CRPC marks a pivotal and challenging phase in prostate cancer treatment. Despite reduced androgen levels, cancer cells adapt by upregulating androgen receptor signaling, activating alternative growth pathways, and increasing intra-tumoral androgen synthesis. This adaptability underscores the malignancy’s resilience and necessitates alternative therapeutic strategies beyond traditional hormone deprivation [10,11].

The management of mCRPC has undergone remarkable evolutions due to enhanced comprehension of the disease’s progression, mutational characteristics, signaling pathways, and mechanisms of drug resistance. The main pathways and mechanisms of action of the principal prostate cancer therapeutic agents are summarized in Table 1 and Figure 1. Abiraterone acetate and enzalutamide extend survival by inhibiting androgen receptor pathways. Docetaxel and cabazitaxel remain integral for patients progressing on androgen receptor-targeted therapies. Sipuleucel-T, an autologous cellular immunotherapy, has shown benefits in select patients, emphasizing the potential of harnessing the immune system against mCRPC [5,7,8]. Radium-223, an alpha-emitter, effectively targets bone metastases, providing palliative benefits and survival advantages by selectively delivering radiation to bone lesions while sparing surrounding tissues. These therapies, often combined, reflect a paradigm shift toward more personalized and multimodal treatment approaches. Ongoing research aims to optimize their use, manage resistance, and improve the quality of life of mCRPC patients, highlighting the dynamic and evolving nature of prostate cancer management [12,13,14].

Nowadays, androgen receptor-signaling inhibitors, such as abiraterone and enzalutamide, and docetaxel are widely used upfront against metastatic castration-sensitive prostate cancer (mCSPC) in combination with ADT. Moreover, the triplet therapy comprising docetaxel, ADT, and inhibitors, such as abiraterone and enzalutamide, has emerged for the treatment of mCSPC. However, cross-resistance between these treatments may occur, reducing the effectiveness of downstream therapies for advanced prostate cancer, particularly mCRPC, giving rise to a more aggressive, treatment-resistant disease phenotype. Therefore, the optimal treatment sequence must be considered. The sequential administration of abiraterone and enzalutamide has been studied and associated with limited efficacy. Nevertheless, cabazitaxel has shown to be effective for mCRPC patients who were previously treated with docetaxel and had disease progression while receiving treatment. In addition to that, 177Lu-PSMA-617 and PARP inhibitors are emerging as effective therapeutic options and have been clinically applied for the treatment of patients with mCRPC. Radioligand therapy with 177Lu-PSMA-617 is a new, effective class of therapy for patients with advanced PSMA-positive mCRPC [12,13,14,15,16].

Recent advances in prostate cancer treatments have notably boosted the survival rates of patients with mCRPC in numerous phase 3 clinical trials. Despite abundant guidelines, identifying the best practices for everyday clinical use, especially treatment sequence and combination strategies, remains challenging because of diverse recommendations influenced by patient health issues and specific disease characteristics [2,11]. This comprehensive review of the literature offers an overview of current therapies and recent breakthroughs in prostate cancer treatment. It specifically highlights bone-targeting radionuclide treatments, particularly radium-223, exploring their integration with other therapies (in terms of timing, concurrent use, layering, or sequencing) and assessing their effectiveness and safety in mCRPC.

## 2. Current Treatment Landscape for mCRPC

### 2.1. Hormonal Therapies Targeting the Androgen Receptor Axis

Abiraterone acetate and enzalutamide play a pivotal role in targeting the androgen receptor axis in mCRPC. Abiraterone inhibits androgen synthesis, while enzalutamide antagonizes androgen receptor signaling, extending survival and improving the quality of life of patients with mCRPC [25].

### 2.2. Chemotherapy

Docetaxel and cabazitaxel are cornerstone chemotherapeutic agents for mCRPC. Docetaxel, often first-line, extends survival and alleviates symptoms. Cabazitaxel, used after docetaxel, offers continued disease control in patients with progressing mCRPC, enhancing overall treatment outcomes [26,27].

### 2.3. Immunotherapy

Sipuleucel-T is a pioneering immunotherapeutic option for mCRPC. This autologous cellular immunotherapy harnesses the patient’s immune system to target prostate cancer cells, offering survival benefits and demonstrating the potential of immunomodulatory approaches to advance treatment outcomes of mCRPC [28,29,30].

### 2.4. Bone-Targeting Therapy

Bisphosphonates, e.g., zoledronic acid, and denosumab, a RANK ligand inhibitor, are integral to managing bone metastases in mCRPC. These agents reduce skeletal-related events, alleviate bone pain, and enhance quality of life, addressing the critical complications associated with bone metastases in prostate cancer patients [31].

### 2.5. Novel Treatments

Emerging therapies for mCRPC include PARP inhibitors, e.g., olaparib, offering targeted DNA damage response inhibition. Prostate-specific membrane antigen (PSMA)-targeted treatment, exemplified by lutetium-177-PSMA-617, shows promise in selectively delivering radiation to PSMA-expressing cells. In addition, targeted alpha therapy, such as radium-223, effectively addresses bone metastases, enhancing treatment options for mCRPC [32,33,34,35].

The main target of radiation is the cell nucleus. Alpha particles, which have a higher relative biological effectiveness than beta particles, deliver extremely high radiotoxicity per particle because of their high linear energy transfer (LET). This property is especially advantageous for treating hypoxic tumors, as these particles have short tissue ranges in addition to high LET, resulting in complex DNA damage that can effectively target radio-resistant cells. Radium-223, the first approved α-emitting bone-targeting radionuclide, has shown benefits in improving overall survival, delaying skeletal-related events, and managing bone pain in mCRPC patients. It uniquely targets osteoblasts, offering a distinctive mechanism of action among life-prolonging treatment options [25,36].

Metastatic prostate cancer cells interact within the bone microenvironment, releasing growth factors that promote osteoblast activity. Radium-223 accumulates within the bone matrix in the vicinity of activated osteoblasts by substituting for Ca^2+^ in hydroxyapatite. Radium-223 behaves like calcium (another alkaline earth metal) in the body, delivering its therapeutic α-particles, which kill cancer cells via radiation-induced DNA damage. Therefore, these α-particles interfere with the positive feedback loop between osteoclasts and osteoblasts that promotes tumor growth (Figure 2) [16].

Radium-223 has characteristics that are typical of alpha particles, such as a short half-life (11.4 days), limited tissue range, and high LET. These features allow it to cause significant damage to DNA with minimal harm to healthy tissue compared with other radionuclides, e.g., beta particles and gamma rays. Its decay process involves the release of four alpha particles, which contribute to 95% of the total particulate decay energy, followed by five additional alpha emissions. It is also highly stable, ensuring minimal distribution of daughter radionuclides. The complete radium-223 decay chain is depicted in Figure 3. If not absorbed into the bones, radium-223 is quickly eliminated through the gastrointestinal tract, with minimal excretion through urine. Only about 20% of the radioactivity remains in the bloodstream 15 min after injection, decreasing to 4% after four hours, while bone radioactivity remains relatively high at 44–77% four hours after injection [37,38].

## 3. Radium-223 in Clinical Trials

### 3.1. Mechanism of Action

Radium-223 operates through a unique mechanism that exploits the bone-seeking properties of alpha-emitting radionuclides. As a calcium mimetic, radium-223 binds preferentially to areas of increased bone turnover, particularly in osteoblastic metastases characteristic of mCRPC. Emitting high-energy alpha particles, it selectively targets cancer cells in the bone microenvironment, inducing localized DNA damage and ultimately leading to cell death. Its short range of penetration limits damage to surrounding healthy tissues. By addressing both the symptoms and underlying pathology of bone metastases, radium-223 offers a targeted and efficacious therapeutic approach for patients with mCRPC, contributing to improved outcomes and enhanced quality of life [37,39].

### 3.2. Clinical Trials and Evidence Supporting the Use of Radium-22

Clinical trials have underscored the efficacy and safety of radium-223 in treating mCRPC. The pivotal ALSYMPCA trial showed significantly improved overall survival and delay in symptomatic skeletal events with radium-223 compared with a placebo [40]. Subsequent real-world studies and meta-analyses corroborated these findings, affirming radium-223’s role as a standard of care in mCRPC management. Moreover, ongoing trials explore its potential with other agents, aiming to further optimize treatment outcomes [19]. The robust clinical evidence supporting radium-223’s use underscores its value as a therapeutic cornerstone of approaches to treating bone metastases and improving survival of patients with mCRPC.

### 3.3. Safety and Efficacy Considerations

Radium-223 has a favorable safety profile, with manageable adverse effects primarily related to hematologic parameters. However, it should be used with caution in combination therapies, as interactions and additive toxicities could arise. Contraindications include prior external beam radiation therapy to the bone due to an increased risk of fracture and significant bone marrow compromise [20]. Adverse effects such as myelosuppression, particularly thrombocytopenia, necessitate careful monitoring and dose adjustments. Despite these considerations, radium-223’s efficacy in prolonging overall survival and delaying skeletal-related events outweighs its associated risks, positioning it as a valuable therapeutic option in the multidisciplinary management of mCRPC [41].

Recent data from ongoing clinical trials offer valuable insights into the management of mCRPC and the potential side effects associated with radium-223. For instance, preliminary results from a phase II trial investigating the combination of radium-223 and the anti-PD-1 antibody pembrolizumab suggest manageable toxicity profiles and evidence of antitumor immune responses. However, common side effects of radium-223 therapy include hematologic abnormalities such as thrombocytopenia and neutropenia, as well as gastrointestinal symptoms like nausea and diarrhea [42]. Such conditions are typically managed through supportive care measures, dose modifications, or treatment interruptions [5,11]. Ongoing efforts focus on optimizing patient selection, treatment sequencing, and dosing regimens to minimize adverse events while maximizing therapeutic benefits in clinical practice.

## 4. Combination Therapies Involving Radium-223

### 4.1. Radium-223 with Hormonal Therapy

mCRPC presents a formidable clinical challenge, necessitating innovative treatment approaches to improve outcomes and the quality of life of patients. Radium-223 has emerged as a promising therapeutic agent in this context, particularly for its efficacy in addressing bone metastases. Hormonal therapies targeting the androgen receptor axis, e.g., abiraterone acetate and enzalutamide, represent another cornerstone of mCRPC management. Combining radium-223 with hormonal therapy holds substantial potential for synergistic effects, enhanced disease control, and improved patient outcomes [43,44].

The rationale for combining radium-223 and hormonal therapy stems from their complementary mechanisms of action and the multifaceted nature of mCRPC. Radium-223 selectively targets bone metastases, delivering localized radiation to cancer cells and minimizing damage to surrounding tissue. Clinical trials have established its efficacy in relieving bone pain and prolonging survival. In contrast, hormonal therapies disrupt androgen receptor signaling pathways, inhibiting tumor growth and progression. However, the emergence of resistance underscores the need for alternative strategies to potentiate their effects [14].

Preclinical studies suggest potential synergies between radium-223 and hormonal therapy. Radium-223’s radiation-induced DNA damage may sensitize tumor cells to hormonal treatment, enhancing their susceptibility to androgen deprivation. Conversely, hormonal therapy may mitigate the emergence of radium-223 resistance by suppressing androgen receptor-mediated survival mechanisms. Furthermore, the bone-targeting nature of radium-223 may counteract the osteoblastic response induced by hormonal therapy, mitigating skeletal complications and preserving bone health [30].

Clinical evidence supporting the radium-223 and hormonal therapy combination is primarily derived from observational studies and retrospective analyses. While prospective trials specifically investigating this combination are limited, subgroup analyses from more extensive trials provide valuable insights. For instance, the ERA 223 trial evaluated the combination of radium-223, abiraterone acetate, and prednisone for patients with asymptomatic or mildly symptomatic mCRPC. Although the trial did not meet its primary endpoint of improved symptomatic skeletal event-free survival, subgroup analyses suggested potential benefits for patients with less advanced disease [45]. Real-world studies have reported encouraging outcomes for the radium-223 and hormonal therapy combination. Shore et al.’s retrospective analysis demonstrated prolonged overall survival and delayed time to symptomatic progression in patients receiving radium-223 in combination with either abiraterone acetate or enzalutamide compared to historical controls. These findings underscore the feasibility and potential efficacy of combining radium-223 with hormonal therapy in routine clinical practice [46]. Despite promising evidence, several considerations warrant attention when employing the radium-223 and hormonal therapy combination. Careful patient selection based on disease stage, symptom burden, and treatment goals is crucial to maximize benefits and minimize risks. Optimal sequencing, dosing, and treatment duration require further elucidation through prospective trials and personalized treatment algorithms [45].

### 4.2. Radium-223 with Chemotherapy

Chemotherapy, particularly docetaxel and cabazitaxel, remains integral to mCRPC management by offering systemic disease control and palliative benefits. Combining radium-223 with chemotherapy is a compelling strategy to address both the skeletal and extra-skeletal manifestations of mCRPC and enhance treatment efficacy. Chemotherapy exerts systemic cytotoxic effects by inhibiting tumor growth and metastatic spread. By targeting both the skeletal and extra-skeletal disease burden, the combination of radium-223 and chemotherapy offers a comprehensive approach to mCRPC management [47].

Preclinical studies support the potential synergies between radium-223 and chemotherapy. Radium-223’s radiation-induced DNA damage may sensitize tumor cells to chemotherapy, thereby enhancing their susceptibility to cytotoxic agents. Conversely, chemotherapy-induced cytotoxicity may potentiate radium-223’s effects by disrupting DNA repair mechanisms and enhancing radiation-induced cell death. Furthermore, the non-overlapping toxicity profiles of radium-223 and chemotherapy allow for safe and tolerable combination regimens that minimize treatment-related adverse events [34,48].

Clinical evidence supporting the combination of radium-223 and chemotherapy is predominantly derived from retrospective and subgroup analyses of more extensive trials [49]. While dedicated prospective trials investigating this combination are limited, observational studies provide valuable insights into its feasibility and potential efficacy. For example, in a randomized phase I/IIa trial, Morris et al. evaluated the radium-223 and docetaxel combination in patients with mCRPC and showed that the combination was well tolerated compared with docetaxel alone [30]. Based on these results, a phase III trial is ongoing to determine the clinical benefit of docetaxel versus docetaxel and radium-223 in mCRPC patients (NCT03574571). These findings underscore the feasibility and potential efficacy of combining radium-223 with chemotherapy for select patient populations. Patient selection based on disease characteristics, treatment history, and performance status is crucial to maximize benefits and minimize risks [20,39,40,41,42,43].

### 4.3. Radium-223 with External Beam Radiotherapy

External beam radiotherapy (EBRT) is a well-established modality in prostate cancer management, offering localized tumor control and palliation of symptoms. The combination of radium-223, which selectively targets osteoblastic bone metastases, and EBRT, which provides precise delivery of high-dose radiation to localized tumor sites and achieves tumor shrinkage and symptom relief, facilitates synergistic targeting of both the systemic and the local manifestations of mCRPC. Therefore, this combination represents a rational strategy for optimizing disease control, mitigating skeletal complications, and enhancing treatment efficacy in mCRPC [4,11].

Preclinical studies support potential synergies between radium-223 and EBRT. The combination treatment has been shown to enhance antitumor effects, induce tumor regression, and prolong survival in prostate cancer bone metastases. Moreover, the bone-seeking properties of radium-223 may augment the therapeutic efficacy of EBRT by sensitizing tumor cells to radiation-induced DNA damage and overcoming radioresistance mechanisms. In addition, the non-overlapping toxicity profiles of radium-223 and EBRT allow for safe and tolerable combination regimens, minimizing treatment-related adverse events [18,31,32].

Clinical evidence supporting the combination of radium-223 and EBRT is primarily derived from retrospective, observational, and subgroup analyses of more extensive trials. While dedicated prospective trials investigating this combination are limited, real-world data provide valuable insights into its feasibility and potential benefits [18,33,34]. The data in the literature have empirically proven the efficacy of combining radium-223 with EBRT, suggesting potential synergistic effects of the combination [40]. Optimal timing, sequencing, and dosing of EBRT relative to radium-223 require further elucidation to maximize therapeutic benefits and minimize potential toxicities. In addition, patient selection based on tumor burden, symptomatology, and treatment goals is crucial to identify those most likely to benefit from this combination therapy.

### 4.4. Radium-223 with Immunotherapy

Immunotherapy, particularly checkpoint inhibitors, has revolutionized cancer treatment by harnessing the immune system to recognize and eradicate tumor cells. The combination of radium-223 and immunotherapy represents a promising strategy to enhance antitumor immune responses, augment disease control, and improve treatment outcomes in mCRPC. During radium-223’s induction of immunogenic cell death by selective targeting of bone metastases and delivery of localized radiation to cancer cells, tumor-associated antigens and danger signals are released, potentially priming the immune system to recognize and attack tumor cells. Immunotherapy enhances antitumor immune responses by blocking immune checkpoint pathways, such as PD-1/PD-L1 or CTLA-4, thereby overcoming immune evasion mechanisms employed by tumor cells [17,36,37].

Preclinical studies support the potential synergies between radium-223 and immunotherapy. Radium-223-induced immunogenic cell death may stimulate dendritic cell activation, T-cell priming, and tumor-specific immune responses. These effects may, in turn, augment the efficacy of immunotherapy by enhancing T-cell infiltration into the tumor microenvironment and overcoming immunosuppressive barriers. Conversely, immunotherapy may potentiate radium-223’s effects by enhancing immune-mediated tumor cell killing and preventing immune escape [17,38].

Preclinical studies of animal models of prostate cancer have demonstrated enhanced antitumor immune responses and improved survival outcomes with the combination of radium-223 and immune checkpoint inhibitors compared with monotherapy. In addition, studies have shown an increase in tumor-infiltrating lymphocytes, upregulation of pro-inflammatory cytokines, and inhibition of immunosuppressive myeloid-derived suppressor cells following combination treatment [39,40].

Despite encouraging preclinical data, clinical trials that specifically evaluated the radium-223 and immunotherapy combination in mCRPC are limited. However, ongoing trials are investigating this combination against various malignancies, including prostate cancer. The combination of radium-223 and pembrolizumab for patients with mCRPC is currently under investigation in a randomized phase II trial (NCT03093428). Preliminary results from this trial have shown promising activity, with manageable toxicity profiles and evidence of antitumor immune responses [35]. When exploring this combination, various aspects, such as patient selection based on tumor characteristics, immune status, and treatment history, should be considered to identify the mCRPC patients most likely to benefit from it. In addition, prospective trials and personalized treatment algorithms are required to clarify optimal sequencing, dosing, and treatment duration.

### 4.5. Radium-223 with Bone-Protecting Agents

The skeletal metastases characterizing mCRPC contribute significantly to morbidity and mortality. Given its bone-targeting mechanism of action, combining radium-223 with bone-protecting agents (BPAs) is a rational approach to the mitigation of skeletal complications, optimizing treatment outcomes, and improving the quality of life of patients with mCRPC [4,8].

The complementary mechanisms of action and the shared goal of preserving bone health in mCRPC patients are the rationale for combining radium-223 with BPAs. Despite its therapeutic benefits, radium-223 treatment may transiently increase bone turnover and the risk of skeletal-related events, particularly in patients with pre-existing bone metastases. BPAs, such as bisphosphonates and denosumab, inhibit osteoclast-mediated bone resorption and reduce the risk of skeletal complications, including fractures, spinal cord compression, and hypercalcemia [8]. The combination treatments with radium-223 and BPAs have been shown to reduce tumor-induced bone destruction, preserve bone integrity, and improve skeletal outcomes. Moreover, BPAs may enhance the therapeutic efficacy of radium-223 by mitigating bone-related toxicities and enabling uninterrupted treatment delivery [5,41].

Clinical evidence supporting the combination of radium-223 with BPAs is derived from observational studies, retrospective analyses, and subgroup analyses of more extensive trials. While dedicated prospective trials investigating this combination are limited, real-world data provide valuable insights into its feasibility and potential benefits. Current evidence demonstrates a significant reduction in the incidence of skeletal-related events and improved overall survival compared with historical controls, suggesting potential synergistic effects of the combination [41,42,43].

Treatment with radium-223 has been associated with an increased risk of fractures in several studies. In the ERA 223 phase 3 trial, the use of radium-223 in combination with abiraterone plus prednisolone resulted in an increased bone fracture risk. However, the incidence of fractures was lower in patients who were taking BPAs at baseline than in patients not taking BPAs [11,17]. Although an increased bone fracture risk was also reported in the phase 3 PEACE III RCT, fracture risk was largely eliminated in both treatment groups (radium-223 plus enzalutamid vs. enzalutamide) with preventative use of BPAs [10,44].

Combining radium-223 with chemotherapy, EBRT, immunotherapy, or BPAs is a promising strategy for managing mCRPC. While clinical evidence supports potential benefits for select patient populations, continued research is crucial to optimize treatment sequencing and dosing. Ongoing or upcoming clinical trials, including the CheckRad-223 trial investigating radium-223 with pembrolizumab, are expected to provide significant insights into combination therapies. These trials aim to elucidate efficacy, safety, and optimal treatment approaches to guide future management of mCRPC.

## 5. Ongoing Research and Future Directions

### 5.1. Investigating Novel Combinations and Sequencing Strategies

As the landscape of mCRPC management continues to evolve, ongoing research is focused on exploring novel combination therapies and sequencing strategies to further optimize treatment outcomes and address therapeutic challenges. This section highlights some key areas of investigation in this dynamic field.

One avenue of exploration involves investigating novel combination therapies that synergistically target different pathways implicated in mCRPC progression. Combinations of radium-223 and other systemic agents, e.g., chemotherapy, immunotherapy, hormonal therapy, or targeted agents, are being actively explored in preclinical models and clinical trials. These combinations aim to capitalize on the complementary mechanisms of action of each agent that potentially enhance antitumor efficacy, delay disease progression, and improve patient outcomes. For example, the combination of radium-223 and enzalutamide has demonstrated safety and promising efficacy in mCRPC, supporting further evaluation of this regimen in the ongoing PEACE III trial [11,22]. Optimization of treatment sequencing is critical for maximizing therapeutic benefits and minimizing toxicities in mCRPC. Understanding the optimal timing and order of administration of different therapeutic modalities, e.g., radium-223, chemotherapy, hormonal therapy, and immunotherapy, is essential for developing personalized treatment approaches. Sequential or alternating treatment strategies may be explored to capitalize on the different mechanisms of action and minimize overlapping toxicities [5].

Several ongoing clinical trials are investigating radium-233 in the context of other therapies for the treatment of mCRPC patients with respect to the effectiveness and safety, timing, and concurrent, layered, or sequential use of such treatment approaches. Several randomized clinical trials are currently underway [PEACE III (phase III, radium-223 plus enzalutamide vs. enzalutamide alone), AlphaBet (phase I/II radium-223 plus lutetium-177 PSMA-I&T), Rad2Nivo (phase II radium-223 plus nivolumab), RADIANT (phase 4, radium-223 vs. ARPI), and DORA (phase III, radium-223 plus docetaxel vs. docetaxel alone)] [45,46,47,48,49], and real-world evidence studies [50,51] exploring radium-223 in mCRPC.

The field of radioligand therapies (RLTs) for the treatment of patients with mCRPC has experienced significant growth over the past decade. Radium-223 and 177Lu-PSMA-617 are currently approved for the treatment of mCRPC patients. Radium-223 and 177Lu-PSMA-617 both prolong OS in different mCRPC settings. The most recent drug approved for beta-particle radiation in prostate cancer is 177Lu-PSMA-617, which has shown clinical benefits in improving OS in mCRPC patients. It uniquely targets prostate cancer cells and the surrounding microenvironment while sparing most normal tissues, offering a distinctive mechanism of action among life-prolonging treatment options. This targeted radioligand therapy has been associated with encouraging biochemical and reduced pain, radiographic response rates, and low toxicity in multiple early-phase studies. VISION is a prospective, open-label, randomized, international, phase 3 trial of targeted radioligand therapy investigating the efficacy and safety of 177Lu-PSMA-617 in a specific population of previously treated mCRPC patients who were selected for PSMA positivity based on PSMA PET imaging. In that trial, radioligand therapy with 177Lu-PSMA-617 plus protocol-permitted standard significantly extended OS among mCRPC patients. In addition to that, RALU is a prospective observational study of radium-223/177Lu-PSMA therapy, investigating the feasibility and clinical outcomes of sequential α- and β-emitter (radium-223 and 177Lu-PSMA therapy) use in mCRPC patients with bone metastasis. The study reported that radium-223 use before 177Lu-PSMA is clinically feasible and well tolerated. Therefore, such sequential treatment approaches should be considered for future assessment of the optimal treatment sequence [14,51]. Actinium-225 is an emerging radioligand therapy, demonstrating promising efficacy and safety outcomes in heavily pre-treated settings. Additionally, novel RLT combinations are emerging to overcome the resistance mechanisms of current RLTs [15].

Another area of interest is the development of biomarker-guided treatment strategies to identify patients who are most likely to benefit from specific therapies. In mCRPC management, the exploration of specific biomarkers holds promise for guiding treatment decisions and improving patient outcomes. Biomarkers that predict treatment response, such as circulating tumor cells, circulating tumor DNA, and molecular signatures, are currently under investigation for their potential utility in predicting treatment response and resistance [52]. These biomarkers offer insights into tumor biology and may help tailor treatment strategies for individual patients, thereby enhancing therapeutic efficacy. However, challenges persist in implementing new combination or sequencing strategies in clinical practice. Optimal biomarker validation, standardization of testing protocols, and integration into routine clinical workflows are essential hurdles to overcome. In addition, the dynamic nature of mCRPC and the heterogeneity in patient populations pose challenges to the identification of universally applicable biomarkers and the development of personalized treatment algorithms. Nonetheless, ongoing research aims to address these limitations, paving the way for more precise and effective management strategies for mCRPC [52,53].

### 5.2. Addressing Unanswered Questions and Optimizing Treatment Approaches

While significant progress has been made in the management of mCRPC, several unanswered questions remain. Ongoing research aims to address these knowledge gaps to optimize treatment approaches and improve patient outcomes. A critical area of investigation is understanding the underlying mechanisms of treatment resistance in mCRPC. Despite initial responses to therapy, many patients eventually develop resistance, leading to disease progression and therapeutic failure. The elucidation of the molecular pathways and tumor microenvironmental factors driving resistance to current therapies, including radium-223, chemotherapy, hormonal therapy, and immunotherapy, is essential for developing novel therapeutic strategies to overcome resistance and enhance treatment efficacy.

Another key research focus is optimizing patient selection and stratification for specific treatments. Identifying biomarkers predictive of treatment response and resistance can help guide treatment decisions and personalize therapy for individual patients. Biomarkers such as genetic mutations, gene expression profiles, and imaging characteristics may provide valuable insights into tumor biology and the treatment response of individuals, enabling more precise and effective therapeutic interventions.

Minimizing treatment toxicity and preserving the patient’s quality of life are paramount considerations in the management of mCRPC. While many treatments offer therapeutic benefits, they may also be associated with significant side effects and adverse events that impact patients’ well-being. Research efforts are underway to develop strategies to mitigate treatment-related toxicities, optimize supportive care measures, and enhance drug tolerability, and adherence to treatment regimens.

Recommendations

Investigate the long-term efficacy and safety of radium-223 in combination with other therapies through dedicated prospective clinical trials, focusing on diverse patient populations and treatment settings.Explore the potential of emerging biomarkers, such as circulating tumor DNA, and molecular imaging techniques to predict treatment response and guide personalized therapeutic approaches in mCRPC.Address challenges in implementing and optimizing new combination or sequencing strategies in clinical practice, including the development of standardized protocols, validation of biomarkers, and integration into routine care pathways.Investigate the economic implications, including cost-effectiveness and healthcare resource use, of new treatment strategies for mCRPC to inform decision-making and resource allocation.Conduct patient-centered research to understand the impact of combination therapies on quality of life, treatment adherence, and survivorship outcomes by incorporating patient-reported outcomes and real-world evidence into study designs.

## 6. Conclusions

The treatment of mCRPC has seen significant progress since the approval of radium-223 a decade ago. The ALSYMPCA trial demonstrated radium-223’s effectiveness in improving overall survival and delaying skeletal-related events in this patient population. Current clinical trials are exploring novel combination therapies and sequencing strategies to optimize the therapeutic benefits of radium-223. This integration represents a significant advance in the field and offers a therapeutic option targeting skeletal manifestations in mCRPC. The pursuit of personalized treatments continues, driven by ongoing research and the quest for patient-centered care.

## Figures and Tables

**Figure 1 diagnostics-14-02407-f001:**
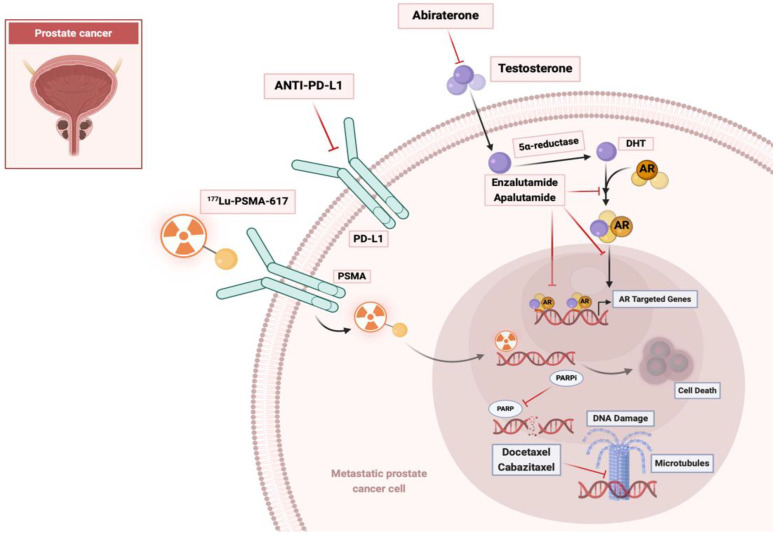
Main mechanisms of therapeutic agents for prostate cancer. Image created with https://www.biorender.com (accessed on 28 October 2024).

**Figure 2 diagnostics-14-02407-f002:**
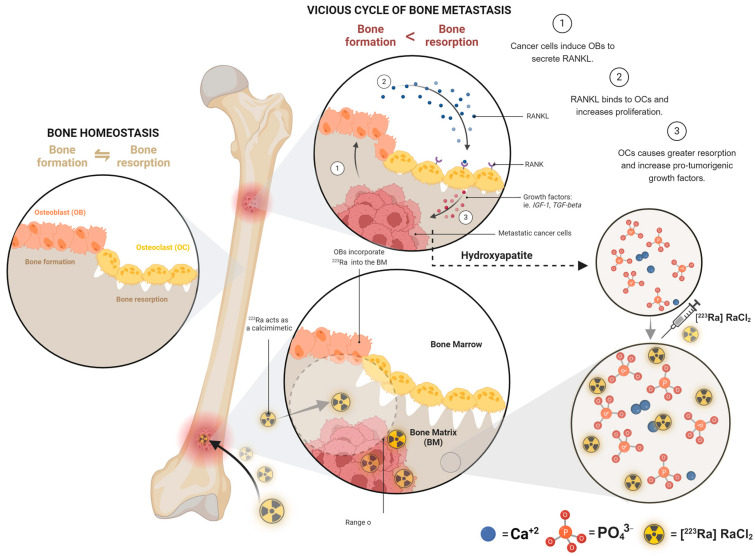
Representation of the vicious cycle of bone metastases and the distribution of radium-223 in the bone ([^223^Ra] RaCl_2_’s mechanism of action). Image created with https://www.biorender.com.

**Figure 3 diagnostics-14-02407-f003:**
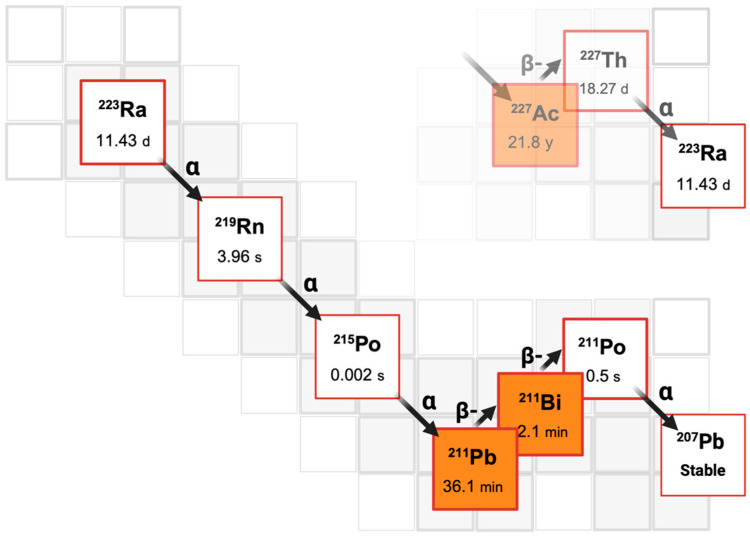
Origin of decay chain-generating radium-223 (^223^Ra) (t½ = 11.43 days) and its subsequent transformations (four α emissions and two β^−^ emissions) to stable ^207^Pb. Image created with https://www.biorender.com.

**Table 1 diagnostics-14-02407-t001:** Drugs and their mechanisms of action against prostate cancer.

Drug	Action	Mechanism
Abiraterone	Inhibition of androgen synthesis	Inhibits CYP17, reduces androgen production [15]
Enzalutamide	Antagonization of androgen action	Androgen receptor inhibitor blocks testosterone effects [16]
Bicalutamide		Blockade of AR [17]
Apalutamide		Prevents AR translocation, DNA binding, and AR-mediated transcription [18]
Docetaxel	Inhibition of mitosis	Tubulin inhibition [19]
Cabazitaxel		
Radium-223	Alpha radiation, gamma rays	Targets bone metastases, emits alpha particles [20]
^177^Lu-PSMA-617	Inhibition of growth signals	Binding and internalization of PSMA ligands triggers cell death [21]
MEDI3726		
Ipilimumab	Checkpoint (CTLA-4) inhibitor	Increases antitumor T-cell responses [22]
Pembrolizumab	PD-1 inhibitor	Regulates T-cell activation [23]
Sipuleucel-T	Immunotherapy	Autologous vaccine [24]

## Data Availability

Data will be made available upon request.

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
