# Peer review of "Bone-Targeting Radionuclides in the Treatment of Metastatic Castration-Resistant Prostate Cancer: A Review on Radium-223 Chloride (Alpharadin) in Combination with Other Therapies"

_diagnostics, 2024, doi:10.3390/diagnostics14212407_

Round 1
Reviewer 1 Report
Comments and Suggestions for Authors
The manuscript is about treatment strategies for metastatic castration-resistant prostate cancer (mCRPC) with radium-223 chlorides (alpharadin). It summaries known treatment options for mCRPC at the beginning. For this purpose, the manuscript contains a summary table and a figure that shows the cellular mechanism of the effect of the drugs. The interaction of radium-223 with the bone environment or the bone metastases and the radioactive decay are then explained. The most interesting are the following overviews of the combination therapies and clinical studies with radium-223. This review ends with an outlook on the future and recommendations for therapy with radium-223.
The review of Alshehri is very well written. ThematicallyIt is very well structured, understandable and informative. It is written as a text book for students and initially does not go into very detail, which is also understandable. The manuscript includes numerous important, high-ranking publications e.g. from journals of NEJM, Lancet Oncology and others, which underlines the quality of this review. Its a pleasure to give my approval for the publication here.
Author Response
Comments: [The manuscript is about treatment strategies for metastatic castration-resistant prostate cancer (mCRPC) with radium-223 chlorides (alpharadin). It summaries known treatment options for mCRPC at the beginning. For this purpose, the manuscript contains a summary table and a figure that shows the cellular mechanism of the effect of the drugs. The interaction of radium-223 with the bone environment or the bone metastases and the radioactive decay are then explained. The most interesting are the following overviews of the combination therapies and clinical studies with radium-223. This review ends with an outlook on the future and recommendations for therapy with radium-223. The review of Alshehri is very well written. Thematically, It is very well structured, understandable and informative. It is written as a text book for students and initially does not go into very detail, which is also understandable. The manuscript includes numerous important, high-ranking publications e.g. from journals of NEJM, Lancet Oncology and others, which underlines the quality of this review. Its a pleasure to give my approval for the publication here.]
Response 1: [I hope this message finds you well. Thank you very much for taking the time to review this manuscript. I appreciate your thorough review and comments.]
Reviewer 2 Report
Comments and Suggestions for Authors
The authors might expand on the use of other radioligands (eg alternating Ra223 with Lu177, Ac223) in CRPC
Author Response
I hope this message finds you well. Thank you very much for taking the time to review this manuscript. I appreciate your thorough review and the opportunity to make the necessary revisions. In response to your requests, I have revised the manuscript accordingly. Below is how I addressed and manuscript each item. Please find the detailed responses below.
Comments: [The authors might expand on the use of other radioligands (eg alternating Ra223 with Lu177, Ac223) in CRPC.]
Response: [The field of radioligand therapies (RLTs) for the treatment of patients with mCRPC have experienced significant growth over the past decade. Radium-223 and 177Lu-PSMA-617 are currently approved for the treatment of mCRPC patients. Radium-223 and 177Lu-PSMA-617 both prolong OS in different mCRPC settings. 177Lu-PSMA-617, the most recent drug approved beta-particle radiation in prostate cancer has shown clinical benefits in improving OS in mCRPC patients. It uniquely targets prostate cancer cells and the surrounding microenvironment while sparing most normal tissues, offering a distinctive mechanism of action among life-prolonging treatment options. This targeted radioligand therapy has been associated with encouraging biochemical and reduced pain, radiographic response rates, and low toxicity in multiple early-phase studies. VISION is a prospective, open-label, randomized, international, phase 3 trial of targeted radioligand therapy investigating the efficacy and safety of 177Lu-PSMA-617 in a specific population of previously treated mCRPC patients who were selected for PSMA positivity on the basis of PSMA PET imaging. In this trial, radioligand therapy with 177Lu-PSMA-617 plus protocol-permitted standard significantly extended OS in among mCRPC patients. In addition to that RALU is a prospective observational study of radium-223/177Lu-PSMA therapy, investigating the feasibility and clinical outcomes of sequential α- and β-emitter (radium-223 and 177Lu-PSMA therapy) use in mCRPC patients with bone metastasis. This study reported that radium-223 use before 177Lu-PSMA is clinically feasible and well tolerated. Therefore, such sequential treatment approaches should be considered for future assessment of the optimal treatment sequence (14,52). Actinium-225 is an emerging radioligand therapy, demonstrating promising efficacy and safety outcomes in heavily pre-treated settings. Additionally, novel RLTs combinations are emerging to overcome resistance mechanisms of current RLTs (15).]
Thank you for pointing this out. I agree with this comment. Therefore, I have expanded on the use radioligand therapy for Metastatic Castrate-Resistant Prostate Cancer including radium-223, 177Lu-PSMA-617, actinium-225. Changes can be found in the revised manuscript page number: 16, 2nd paragraph, and 582.]
Reviewer 3 Report
Comments and Suggestions for Authors
The work rapresents a good summary of the state of the art regarding Radium.
I would organize the 'Overview of current treatments and recent advances' part better, explaining the possible therapies better and underlining the possible therapeutic sequences. to update the table. better propose the role in which line of therapy Radium can be used.
It would be necessary to underline more any differences and similarities compared to other radiologists (e.g. Lutezio).
Comments on the Quality of English LanguageGood English language
Author Response
I hope this message finds you well. Thank you very much for taking the time to review this manuscript. I appreciate your thorough review and the opportunity to make the necessary revisions. In response to your requests, I have revised the manuscript accordingly. Below is how I addressed and manuscript each item. Please find the detailed responses below.
Comments: [The work rapresents a good summary of the state of the art regarding Radium. I would organize the 'Overview of current treatments and recent advances' part better, explaining the possible therapies better and underlining the possible therapeutic sequences. to update the table. better propose the role in which line of therapy Radium can be used. It would be necessary to underline more any differences and similarities compared to other radiologists (e.g. Lutezio).]
Response: [Nowadays, androgen receptor-signaling inhibitors, such as abiraterone and enzalutamide, and docetaxel are widely used upfront against metastatic castration-sensitive prostate cancer (mCSPC) in combination with ADT. Moreover, the triplet therapy comprising docetaxel, ADT, and inhibitors, such as abiraterone and enzalutamide has emerged for the treatment for mCSPC. However, cross-resistance between these treatments may occur reducing the effectiveness of downstream therapies for Advanced prostate cancer, particularly mCRPC, giving rise to a more aggressive, treatment-resistant disease phenotype. Therefore, the optimal treatment sequence must be considered. The sequential administration of abiraterone and enzalutamide has been studied and is associated with limited efficacy. Nevertheless, cabazitaxel has shown to be effective for mCRPC patients who were previously treated with docetaxel and had disease progression while receiving treatment. In addition to that 177Lu-PSMA-617 and PARP inhibitors are emerging as effective therapeutic options, have been clinically applied for the treatment of patients with mCRPC. Radioligand therapy with 177Lu-PSMA-617 is a new effective class of therapy for patients with advanced PSMA-positive mCRPC (12-16).]
Thank you for pointing this out. I agree with this comment. Therefore, I have explained the possible therapies for metastatic castrate-resistant prostate cancer as suggested and underlined the possible therapeutic sequences. Changes can be found in the revised manuscript page number: 3 and 4, 4th paragraph, and 120.] Due to the heterogenous nature of mCRPC, patients may require multiple lines of therapy. However, determining optimal treatment sequences to ensure patients derive the best OS benefit while maintaining quality of life remains challenging. References to the line of treatments are referenced and explained in details in the text under each treatment section "combination therapies involving radium-223", they can be found in the revised manuscript starting from page number: 4, 2nd paragraph, and line 360.] Additionally, I have covered the use of radioligand therapies in mCRPCa in particular 177Lu-PSMA-617. Changes can be found in the revised manuscript page number: 16, 2nd paragraph, and 582.]